# Evaluation of Smart Sensors for Subway Electric Motor Escalators through AHP-Gaussian Method

**DOI:** 10.3390/s23084131

**Published:** 2023-04-20

**Authors:** Ruan Carlos Alves Pereira, Orivalde Soares da Silva, Renata Albergaria de Mello Bandeira, Marcos dos Santos, Claudio de Souza Rocha, Cristian dos Santos Castillo, Carlos Francisco Simões Gomes, Daniel Augusto de Moura Pereira, Fernando Martins Muradas

**Affiliations:** 1Military Engineering Institute—IME, Rio de Janeiro 22290-270, Brazil; 2Department of Production Engineering, Faculty of Engineering, Praia Vermelha Campus, Federal Fluminense University, Niteroi 22040-036, Brazil; 3Production Engineering Department, Federal University of Campina Grande (UFCG), Campina Grande 58428-830, Brazil; 4Operational Research Department, Naval Systems Analysis Center (CASNAV), Rio de Janeiro 20091-000, Brazil

**Keywords:** decision making, 4.0 industry, automation, operational research, subway, predictive maintenance, bibliometric and electric motors

## Abstract

This paper proposes the use of the AHP-Gaussian method to support the selection of a smart sensor installation for an electric motor used in an escalator in a subway station. The AHP-Gaussian methodology utilizes the Analytic Hierarchy Process (AHP) framework and is highlighted for its ability to save the decision maker’s cognitive effort in assigning weights to criteria. Seven criteria were defined for the sensor selection: temperature range, vibration range, weight, communication distance, maximum electric power, data traffic speed, and acquisition cost. Four smart sensors were considered as alternatives. The results of the analysis showed that the most appropriate sensor was the ABB Ability smart sensor, which scored the highest in the AHP-Gaussian analysis. In addition, this sensor could detect any abnormalities in the equipment’s operation, enabling timely maintenance and preventing potential failures. The proposed AHP-Gaussian method proved to be an effective approach for selecting a smart sensor for an electric motor used in an escalator in a subway station. The selected sensor was reliable, accurate, and cost-effective, contributing to the safe and efficient operation of the equipment.

## 1. Introduction

The Industrial Revolution marked a turning point in the history of humanity, leading to the development of new technologies and innovations that changed the way we live and work. One of the key advancements of the Industrial Revolution was the development of sensors for machines [1,2,3,4], which helped to increase efficiency and productivity in manufacturing. With the advent of the Fourth Industrial Revolution, or Industry 4.0, the use of sensors has become even more widespread, leading to the creation of smart factories and the integration of advanced technologies such as the Internet of Things (IoT) and artificial intelligence (AI) [1,5].

Scholars and practitioners [6,7,8,9] have started to analyze relationships between Industry 4.0 technologies and environmental management. In the manufacturing sector specifically, some companies are implementing solutions based on smart sensors and AI, especially to improve energy consumption, while others are focusing on additive manufacturing to conserve and reuse resources [3]. Although some companies are implementing Industry 4.0 technologies to address waste management, it is unclear what kind of Industry 4.0 technologies may assist with this or with air and water pollution. On this uncertain ground, authors such as [10] believe that Industry 4.0 technologies may hamper environmental performance.

Sensors play a crucial role in Industry 4.0 by enabling machines to collect and analyze data in real time, which can be used to improve production processes and increase efficiency [11,12,13,14,15,16,17,18,19]. The research of [20] confirms that sensors can perform such roles. According to Refs. [21,22], the sooner the electrical anomaly is identified, the longer the best performance of the electric motor will be maintained. This concept is characterized by bringing the preventive maintenance intervention, by predictive methods, to point P of the P-F curve, as found in Ref. [23] and shown in Figure 1. Thus, we can monitor the performance of machines, detect potential problems, and trigger maintenance activities before they become critical. This proactive approach to industrial maintenance can help by reducing downtime, minimizing the risk of equipment failure, and extending the lifespan of important parts of machines, such as electric motors [21,24,25].

The objective of such predictive maintenance is to reduce maintenance costs and increase the availability of the machine in an attempt to identify various issues in a system from the start of the degradation. Based on this information, the manager can decide which type of maintenance intervention would be better [23].

The Figure 1 show three examples of conditions of machine or system. The first condition, indicated by the green line is a perfect condition of the system, after the green line, the system start to decrease their functions condition. The point “P” represent the increase probability point of potential failure. Without maintenance or any intervention, the system could present a functional failure, generating stops or loss of productivities in the system.

In addition, among the industrial equipment currently used, those that have electric motors as an integral part of the equipment or system are significant in quantity and criticality, allowing machines to perform the required tasks [26]. Electric motors are important devices for many industrial processes, as they are widely used as primary movers of most of the loads involved in these applications [27]. According to Refs. [10,28], their vast usage can represent, in terms of electrical consumption, between 40% and 60% of the total consumption on any industrial site.

Smart sensors can monitor a variety of parameters in an electric motor, such as temperature, vibration, and electric tension, providing valuable insight into its performance and condition [10]. These data can be used to identify potential issues before they escalate, reducing downtime and increasing productivity [29,30].

Additionally, the use of smart sensors can also help companies to optimize their energy usage, as well as track and manage their carbon footprint [7,31]. With this information, companies can make informed decisions about how to reduce their environmental impact while improving their bottom line, as reported by Ref. [31]. However, there are several types of smart sensors available on the market. It is necessary to evaluate which sensor is most suitable for a specific operation, considering physical parameters of the process, such as temperature and vibration range, and aspects such as the financial ability to purchase these sensors [32,33,34].

In this context, this paper focuses on choosing the most suitable smart sensor for monitoring an electric motor through an integrated monitoring system with smart sensors, as shown in Figure 2. For this, we propose an AHP-Gaussian decision support model, which is applied to evaluate the most suitable sensor for an EMOD 440 V CA 19 A 10.5 KW 50 Hz electric motor used in the movement system of an escalator in an important subway station. In addition, we show how the chosen sensor works, graphically displaying the parameters collected by the smart sensor after a period of reading in a real environment of monitoring an electric motor.

There are several types of smart sensors, which vary in terms of aspects such as brand, durability, accuracy, functionality, availability, price, and adaptability to the operation. As there are several smart sensor options for the monitoring of an electric motor, it is necessary to understand which sensor is the best for a given application [10].

Smart sensors can have different levels of accuracy and reliability in measuring electric motor variables, such as temperature, vibration, current and voltage. When choosing a suitable sensor, it is important to consider the precision and reliability needed for the specific application; if the choice is not adequate, the sensor may not collect the data generated by the system completely [2].

Some sensors may not be compatible with the electric motor or monitoring system used. It is important to choose a sensor that is compatible with the monitoring platform and that can be easily integrated into the system [6].

Different sensors can be designed to operate in different environments, such as environments with high humidity, high temperature, or exposure to chemicals. It is important to choose a sensor that can operate safely and reliably under the environmental conditions of a given application [8].

Considering the risk of acquiring and installing a sensor that may not be suitable for an operation, the task of selecting a smart sensor through a decision support methodology is justifiable and relevant, as proposed by this work.

## 2. Smart Sensors Application

Smart sensors can be found in a wide range of applications, from industrial machinery, subway stations, and smart homes to wearable devices and medical equipment. They can measure a variety of parameters, such as temperature, humidity, light, and motion, providing real-time data to help monitor and control systems [35,36].

According to Ref. [37], in the conventional approach, the sheer number of accompanying wires, fiber optic cables, or other physical transmission mediums may be prohibitive, particularly for structures such as long-span bridges or tall buildings. Consequently, global wireless communication technologies, which will facilitate low-cost, densely distributed sensing, have been investigated [38].

A sensor is a device designed to take information from an object and convert it into an electrical signal [38,39]. According to Ref. [40], conventional integrated sensors can be divided into three parts (Figure 3): (i) the sensing element; (ii) signal conditioning and processing (e.g., amplification, linearization, compensation, and filtering); and (iii) sensor interfaces (e.g., wires, plugs, and sockets for communication with other electronic components).

As illustrated in Figure 4, the essential difference between a smart sensor and a standard integrated sensor is its intelligence capabilities, i.e., the onboard microprocessor. The microprocessor is typically used for digital processing, analog-to-digital or frequency-to-code conversions, calculations, and interfacing functions, which can facilitate self-diagnostics, self-identification, or self-adaptation (decision-making) functions [38]. It can also decide when to dump/store data and control when and for how long it will be fully awake to minimize power consumption [6,27].

According to Refs. [7,41,42], smart sensors are devices that combine the traditional functions of sensors with additional capabilities such as data processing, communication, and self-awareness. Some key qualities of smart sensors are presented in Table 1.

The use of smart sensors allows for greater efficiency, improved decision making, and reduced costs. In industrial settings, for instance, smart sensors can monitor equipment and send alerts when maintenance is needed, reducing downtime and increasing productivity [36]. In healthcare, IoT sensors can be used to monitor patients and track their vital signs, improving patient outcomes and reducing healthcare costs [43].

There are several types of smart sensors available on the market from different manufacturers which have types of uses. In this way, the selection of a smart sensor for a given application is crucial, since the multiple functions of a smart sensor may vary according to the type of smart sensor, impacting the effectiveness of use during operation.

The context of the research into and application of smart sensors, as well as the most appropriate choice of sensors for certain applications, is a field of study found in several literature sources. In the next section of this work, a literature review is described in order to identify recurrent and adherent parameters to the application of this work on sensor types, application examples, sensor selection parameters, and other aspects that are adherent to the work.

## 3. Literature Review

The decision-making process of selecting a smart sensor for the application and monitoring of an electric motor has been the focus of several previous studies under different approaches. Papers were identified in databases such as SCOPUS and Science Direct that had final usefulness for the research. These scientific bases were selected due to their relevance in the field of this study.

The literature mapping process was carried out in six steps, in the SCOPUS and Science Direct scientific databases, as described below.

An initial search with the terms “Smart sensor” AND “Selection” in titles, abstracts, and keywords;The limitation of the search to works in English;The limitation of the search with keywords “MCDM”, “Smart sensors”, “Intelligent control”, “Sensors networking”, “Internet Of thing”, “Smart sensor”, “IoT”, “automation”, “Decision support system”, “Maintenance”, and “microsensors”;The removal of duplicate works between the two searched databases;The acquisition of works for complete reading;The complete reading of the works and a search for adherence to the current work.

The research was carried out from 5 April to 6 April 2023 via the internet, with access to the scientific bases of the CAPES BR journal portal, with authorization from a federated academic communication. Access was gained with a login from the IME (Military Institute of Engineering).

Table 2 shows, in quantitative terms, the evolution of the exploratory research process for understanding the theme through works found in the literature. In Table 2, each step is represented according to the six steps mentioned. The first column indicates the step of the search, the second column presents a brief description of this stage, the third column shows the number of works found in the SCOPUS base, and the fourth column shows the number of works found in the ScienceDirect base.

Seeking to create a visual approach to how the articles found and used as the basis of this work were related, Figure 5 was generated through the VOSviewer Online version 1.2.1 software, which represents the way the main terms found in the titles and abstracts of the articles were connected. Each different color indicate a cluster of papers.

It was possible to identify three clusters, composed of (1) smart sensors, sensors, and induction motors; (2) industry and advances, and (3) the future and challenges. These formations of clusters, as well as their connections, show an overview of the approach of articles on the subject. Figure 6 shows a density map of the main themes found. 

In Figure 6, the term smart sensor is the densest, followed by the term sensor. The other terms have similar densities. This density analysis is interesting because it shows the main biases of the articles that make up the base of this research.

Figure 7, generated through the LITMAP software online version 1.0, shows how the most important articles that made up the research base are related, how they are distributed over time, and which are the most cited.

After the inclusion of these works, we defined the bibliographic base of the research, which served as a reference, inspiration, and support for this work. To detail the approach of each article, in the following paragraphs of this section, a brief description of subjects related to the articles is presented.

The research of [44,45,46] clearly describes how it is possible to carry out a decision-making process through the AHP-Gaussian model as well as other aspects related to MCDMs. Still, the MCDM approach and the work of [34] use the AHP and MOORA decision-making models for a hybrid approach to a decision support problem involving smart sensors, in line with the objective intended by the current work.

In a broad view of the literature, articles were found that address the use of technological apparatus and present several applications with smart sensors. The works of [2,10,39,41,42,47] describe a view of the application of smart sensors considering the precepts of Industry 4.0, explaining how to integrate IoT with present concepts from Industries 3.0, 2.0 and 1.0.

To guide the research towards the issues of monitoring electric motors, the works of [26,30] were found, which performed an analytical combination for electric motors through smart sensors, involving transient and stray flux analyses, with the differentiation of current signature analysis in the work of [30], as found in Refs. [21,32,48]. The work of [26] used, as a point of differentiation, a thermoenergetics approach to evaluate the performance of an electric motor.

The relation of the performance analysis of an electric motor through stray flux is a representative type of analysis to monitor the performance of electric motors using smart sensors, which is explored in the works of [28,33,37].

In addition to this approach, other works such as Refs. [20,23,29,49] use parameters such as vibration and temperature, where, integrated with RFID technologies to perform performance analyses, they optimize the use of electric motors.

The analysis and monitoring carried out in electric motors are activities based on the concepts of predictive maintenance, which seek to anticipate equipment failure, reducing costs with breakdowns and periods of unavailability. The works of [22,24,27,38,50] address the relationship between predictive maintenance issues and the use of smart sensors, seeking to keep assets up-to-date and available.

The movement of updates and modifications of the assets of a manufacturing unit, within the context of Industry 4.0 through the use of smart sensors, is addressed by the works of [6,8,51,52]. This approach helps clarify points related to how IoT systems can be integrated with machines and makes the environment more productive, safe, and sustainable.

However, this movement to bring reality closer to concepts and technological devices such as IoT is not an industrial exclusivity. Works such as [3,35,43,53,54,55,56,57] describe the progress and issues related to the use of smart sensors through the smart home concept, which seeks to automate residential functions in search of comfort, security, and user satisfaction. This is highlighted in [43], which, like [36], uses a recognition approach integrating a smart sensor into the sensor’s intelligence system.

For both industrial and residential sensing issues to advance and establish new limits of technological integration, advances in digital architecture and developments in computer science are required. Works such as [58,59,60,61,62,63,64,65,66] report structures, codes, architectures, and other forms of development of network sensors, working together with physical devices.

Another information technology issue that, combined with the use of smart sensors, can bring several advances to society, is the use of the blockchain concept combined with smart sensors [67,68].

During research, the construction segment has also been considered a target of interest in the study of performance evaluation through smart sensors. Articles [1,5], as well [69,70], address ways to use smart sensors for application in the construction industry.

In this way, by intelligently integrating devices, in construction, industrial, or residential environments, it is possible to approach the concept of a smart grid. This concept, associated with the use of smart sensors, is discussed in [9,40,71,72,73,74,75]. The works present methods and examples that relate the use of smart sensors to the concept of and application in smart grids and smart cities.

In an applied way, Refs. [72,76,77] show how smart sensors can have a biological application, seeking developments in the area of health and health care.

Works that addressed the use of smart sensors for logistic applications and environmental performance evaluation were identified. These works, such as [4,7,22,31], show that the use of smart sensors can provide enough information to the decision maker, making them capable of promoting actions that increase the enterprise’s profitability and reduce environmental impacts.

Based on the literature review, several parameters and methodologies that need to be applied to perform the selection of a smart sensor were identified. A sensor’s performance evaluation parameters vary according to its application needs. For cases in which monitoring is carried out in environments with water incidence, it is important to have IP 66 or 69 protection, for example [29]. Other parameters are widely used in the literature, such as temperature range, vibration range, and sensor voltage measurement capacity.

For the application of this work, considering the parameters found in the literature, those that had greater adherence to the project were (1) max. working temperature, (2) max. vibration range, (3) weight, (4) max. distance communication, (5) max. electric power, and (6) max. data traffic speed. In addition to these parameters, (7) acquisition cost was also mentioned as relevant.

The importance of MCDM in selecting a smart sensor lies in its ability to evaluate multiple criteria simultaneously. When selecting a smart sensor, there are often several criteria to consider, such as accuracy, reliability, cost, and compatibility with existing systems. MCDM allows decision makers to prioritize these criteria and compare different options based on their performance against each criterion [32].

Analyzing the decision support methodologies found in the literature review, the AHP-Gaussian method was the method that proved to be the most adherent for application. The main advantage of using the method for this application is the interpretation of data directly from the data sheets and catalogs of the sensors. In this way, biases on choice are avoided during the selection process, and the sensor that obtains the best performance among the chosen parameters is prioritized. This methodology understands that normal events can help in decision making and the formation of weights for criteria in a decision support system [44].

## 4. Methodology

The reason for choosing the AHP-Gaussian method was due to the quantitative aspects of the alternatives and because it significantly reduces the cognitive effort of decision makers, compared to the traditional AHP, which requires paired comparisons with all alternatives [46].

The AHP-Gaussian method is a compensatory method, and the attributes are independent, converting qualitative attributes into quantitative attributes. The method has greater adherence in applications in which it is necessary to obtain a decision between alternatives, but there is no full availability of decision makers to carry out the paired evaluation of alternatives as performed in the original version of AHP.

Another clear objective of the AHP-Gaussian method is that the hierarchy of the alternatives avoids the confirmation bias of the decision makers about various alternatives, only taking the performance of the alternative as a premise, compared to the performance of the other alternatives to each criterion [44]. This aspect was critical in selecting the method to apply in this paper.

The method starts with the basic principle that normal events occur following the normal curve; thus, the probability density function of occurrence of a given performance, in an alternative criterion, follows Gaussian logic, aiming to create a new approach to the original AHP method, which is based on a sensitivity analysis derived from the Gaussian factor [45].

Related to the analysis of complex problems, Multi-Criteria Decision Analysis (MCDA) is a field of Operations Research (OR) that provides the structuring of problem variables, enabling the analysis of complex problems in evaluations of different types, considering risk and uncertainty in a transparent format [78].

The MCDA methods, such as AHP and AHP-Gaussian, are very useful to support the decision-making process in these cases because they consider value judgments, not technical issues alone, to evaluate alternatives to solving real problems, creating a highly multidisciplinary approach [79,80].

The adoption of a combination of methodologies enables the identification of variables and a rational analysis of information. The AHP method is also widely used in conjunction with other MCDA methods [81,82].

The AHP method is one of the classic methods (uncompensated methods) in American schools, developed by [83] to solve quantitatively complex problems with multiple decision criteria. The essence of this method is to weight criteria in a two-by-two comparison using a metric known as the Saaty baseline scale, shown in Table 3.

According to Ref. [44], the Analytical Hierarchy Process (AHP) is a widely used method for decision making and problem solving. It is based on a multi-criteria decision-making model that helps to determine the relative importance of different criteria in a decision-making process. 

The AHP-Gaussian approach can be used in a variety of industries and applications, such as engineering, finance, and healthcare. It can help decision makers to analyze complex problems, evaluate different alternatives, and make informed decisions based on data and the relative importance of different criteria [45].

According to Ref. [83], to ensure consistency in decisions, the parties must use the AHP method to calculate the consistency ratio (*CR*) between the Judgmental Consistency Index (*CI*) and the Random Consistency Index (*RI*). The model has a maximum misfit tolerance of 10%; therefore, *CR* ≤ 0.1. If *CR* > 0.1, the decision maker must carry out a new evaluation of the criterion. Next, we present a summary of the AHP method according to Refs. [84,85].

(a)The formation of the decision matrices (*m* × *n*), Equation (1):


(1)
$$\left[\begin{array}{ccc} a11 & \ldots & a1n\\ \vdots & \ddots & \vdots \\ am1 & \ldots & amn \end{array}\right]$$


(b)The calculation of the eigenvector (*Wi*), Equation (2):


(2)
$${Wi=\left(\prod_{j=1}^{n}Wij\right)}^{1/n}$$


This step consists of ordering the priorities or hierarchies of the studied characteristics.

(c)The calculation of eigenvectors’ normalization, enabling comparability between criteria and alternatives, Equation (3):


(3)
$$T=|\frac{W1}{\sum Wi};\frac{W2}{\sum Wi};\frac{W3}{\sum Wi};\frac{Wn}{\sum Wi}|$$


(d)The determination of the index that relates to the criteria of the consistency matrix and the criteria weights, Equation (4):


(4)
$$\lambda_{max}=T\times W$$


(e)The determination of the consistency index (*CI*), Equation (5):


(5)
$$CI=\frac{\lambda_{max}-n}{\left(n-1\right)}$$


(f)The calculation of the consistency ratio (*CR*), Equation (6):

(6)$$CR=\frac{CI}{RI}$$
where *RI* is the random consistency index obtained for a reciprocal matrix of order *n.* This step allows for the evaluation of the inconsistency due to the order of the judgment matrix. If the *CR* value is greater than 0.10, the decision maker must review the model or judgments.

The Gaussian AHP method proposed by [84] provides a new approach to existing methods based on the sensitivity analysis of the Gaussian factor. In this version of the AHP method, the criteria weights are calculated using the decision matrix itself. The reduction in the decision maker’s cognitive effort is a differentiating factor of this method from other AHP applications. This is because decision makers do not have to spend time reviewing between criteria to obtain each weight later. It is noteworthy that this model is viable in scenarios where the alternatives are quantitative in the analyzed criteria [29].

Figure 8, adapted from Ref. [84], presents a graphical representation of the application steps of the AHP-Gaussian method.

The MCDA methods, such as AHP-Gaussian, are very useful to support the decision-making process in these cases because they consider value judgments, not technical issues alone, to evaluate alternatives to solve real problems, presenting a highly multidisciplinary approach [86].

### Limitation of AHP-Gaussian

The AHP-Gaussian method, as a method that is characterized and used as a compensatory method with independent attributes, has two main limitations:Negative numbers: One limitation of the use of AHP-Gaussian is the impossibility of using parameters that present negative values. This limiting aspect needs to be considered due to the axiomatic aspects present in the model formulation [83].Quantitative evaluation: The AHP-Gaussian method observes data in a strictly quantitative manner. This aspect can put the decision system at risk if the data to be as used performance metrics for the alternatives are not fully representative. An alternative to this limitation is the translation of qualitative aspects into a quantitative scale; for the application of the model, however, care is needed so that the data are representative and do not generate a preference and restriction to a given alternative [86].

These two main limitations of the method are characteristics that are important for specific decision-making problems; evaluating the replacement of the use of the model when these restrictions are part of the structure of the problem is sufficiently important in its application.

## 5. Application—Smart Sensor Selecting

This work concentrates the selection of sensors that are most representative in the global and Brazilian industries, considering applications focused on measuring aspects related to vibration, voltage, power, and temperature [24,29,30].

For the application, interviews were conducted with the entire technical team involved in the maintenance and asset management of the target company of the study. The team included a field manager, an applications engineer, two maintenance technicians, a mechanic, and an electrician. These professionals were consulted to validate the identified parameters and the types of pre-selected sensors as selection alternatives. All professionals were in accordance with the identified parameters and with the sensors chosen as an alternative for selection.

In this way, through bibliographic research and interviews with specialists in the maintenance of electric motors, seven parameters inherent to the sensors were selected that, together, make it technically and operationally feasible to install sensors to monitor an electric motor.

Table 4 presents the evaluation parameters and a description of these parameters. This description was conceived through interviews with professionals in the area of the maintenance of electric motors and a literature review, as found in Refs. [24,29,30]. Table 5 represents, therefore, the criteria against which the alternatives will be evaluated.

The growth of the “data-driven” culture opens space for decision making, machine learning (ML) and other types of data analyses to support decisions [87]. To support the selection of a sensor, compared to several other sensors available on the market, it is necessary to understand the relevant parameters in this multi-criteria evaluation.

The sensors that were listed as alternatives for the work are easily counted and acquired worldwide and have good applicability in the systems outlined for the application of this work. Table 5 presents the main parameters of these smart sensors [88,89,90,91].

All four sensors presented in Table 5 are from the same class of sensors, considering smart multi-attribute sensors, capable of accurately measuring and transmitting physical parameters such as housing temperature, voltage, rotor speed, and vibration in an EMOD 440 V AC 19 A 10.5 KW 50 Hz electric motor. In this way, they were classified as alternatives for the decision work carried out through the AHP-Gaussian multicriteria decision support model. 

As described in Table 3, each sensor was identified by the terms A1, A2, A3, and A4. These identifications helped in the identification of each sensor when evaluated as alternatives by the AHP-Gaussian decision support model.

### 5.1. Smart Sensor Monitoring for Subway Motor Escalator

The goal of this work was to select a sensor to perform the monitoring of an electric motor, EMOD 440 V AC 19 A 10.5 KW, used in an escalator of a large subway station, shown in Figure 9. The blue circle indicate the Electric motor that is been used for this operation.

In a subway station located in the South Zone of Rio de Janeiro, there was a risk of discontinuity in the functioning of the escalators. During the New Year’s Event, there were constant unscheduled stops due to the greater flow of people at the station.

The main causes of failures identified by the experts were mechanical and electrical, causing serious inconvenience to the station. In addition to the inconvenience caused in customer service and the difficulty of emergency repairs, there was the risk of legal proceedings.

There are several important measurement parameters to monitor in an electric motor, such as rotor speed, torque, electrical anomalies, possible bearing failures, or loss of efficiency [49]. To analyze these parameters, the sensors need to have adequate measurement capacity. Otherwise, a physical event may occur in the engine and not be seen by the sensor [34,55,67,77].

For this application, the vibration, temperature and voltage parameters were monitored and evaluated in terms of their form, in search of anomalies that can be corrected in the system. Figure 10 graphically represents how the elements of sensing, measurement, and interface are related.

### 5.2. Application of AHP-Gaussian Methodology 

In this session, we evaluate the sensors obtained in Table 3, which are considered the alternatives for this selection. These A1, A2, A3, and A4 sensors were considered as alternatives due to their global representativeness, availability, ease of access to data, and possibility of integration into existing systems in practical applications.

It is important to mention that any smart sensor capable of monitoring an EMOD 440V AC 19A 10.5 KW electric motor, as described for this application, can be evaluated according to these criteria, which is a global framework for evaluating the feasibility of installing a device.

Table 6 describes the value for each alternative regarding the criteria defined for the application and, performing a correlation with the image that represents the steps of the AHP-Gaussian method, Table 6 and Table 7 represent the first step in the method. The information about the performance of each smart sensor was obtained directly from data sheets [88,89,90,91] and interviews with maintenance professionals.

Table 7 shows the normalized decision matrix, still representing step 1, depicted in Figure 8.

Table 8 describes step 2 in Figure 8, averaging the criteria values for the alternatives. Table 8 also describes step 3, showing the calculation of the standard deviation of the criteria based on the alternatives, and also describes step 4, establishing the Gaussian factor for each of the criteria, obtained by dividing the mean by the standard deviation in each criterion. Then, after obtaining the Gaussian factor, the normalized Gaussian factor is calculated.

This matrix is identified as matrix $A_{i}C_{j}$, where *i* represents each one of the alternatives, and *j* represents the criteria.

Each normalized Gaussian factor is identified by ${GN}_{n}$, where ${GN}_{1}=0.099459$ and represents the normalized Gaussian factor of the C1 criterion. In the same way, ${GN}_{n}$ is obtained for each criterion.

Table 9 describes step 5, weighting the decision matrix through the values found in the normalized Gaussian factor of each criterion, ${GN}_{n}$, multiplied by the normalized performance value in each criterion, for each alternative $A_{i}C_{j}$.

Described in Table 10 are step 6 and 7, referring to Figure 8, representing the sum value, for each alternative, of the product between the normalized performance value by the normalized Gaussian factor, through which it is possible to obtain a normalized index preference for each of the alternatives.

Through this index, it is possible to establish a ranking of the alternatives of smart sensors evaluated by the model.

The AHP-Gaussian model, as its main differential, manages to weigh weights between the criteria, saving the decision maker’s cognitive effort. The use of a decision support model for decision making in situations that will impact costs and influence the life of society is highly recommended [92,93]. In this application, the Smart sensor ABB Ability Sensors were selected based on the described criteria and the performance of the alternatives. This sensor, indicated by the model, was installed in an EMOD 440 V AC 19 A 10.5 KW 50 Hz motor responsible for moving an escalator in a subway station. This monitoring saved corrective maintenance costs and avoided equipment downtime. The following section describes the information obtained by monitoring the smart sensor in the application.

## 6. Results and Discussion

The selected sensor was installed to monitor an EMOD 440 V 19 A 10.5 KW 50 Hz electric motor, responsible for keeping the escalators in operation at an important subway station. In Figure 11, it is possible to observe the sensor indicated by the AHP-Gaussian method, the ABB Ability Smart sensor installed in the monitored motors. The blue circles indicate the smart sensor position, installed at motors.

After installing the sensors in the motors and monitoring periods, it was possible to detect moments of global vibration peaks, frame temperature, and bearing condition, which are occurrences that can reduce the shelf life of motors. After the automatic calls were sent by the sensors to AS3 consultancy, the company responsible for monitoring the assets, the necessary adjustments were made to normalize the operation of the assets.

In Figure 12, it is possible to observe the behavior of motor 1. The motor has a stable behavior, but with some temperature rise peaks in the motor frame.

On 18 December, as indicated on the graph, a maintenance procedure was carried out on motor 1. After this procedure, an increase in global vibration was detected to levels above normal, and a decrease in frame temperature was observed, as shown in Figure 13. This behavior was considered anomalous behavior by the integrated sensor data processing system.

On the same date, the same maintenance procedures were carried out on motor 2. However, the behavior of the motor manifested in an increase in temperature to levels considered within the normal range and an increase in vibration to levels above normal, as shown in Figure 14.

Based on this information and the anomalous behavior of the vibration of motors 1 and 2, the system indicated a possible failure in the motor drive system, relating the rotation speed to the high vibration.

Through this indication, it was verified that until 18 December, the frequency inverter that drives motor 1 and 2 had a defect and, therefore, caused a reduction in the motor operating speed. After its correction, the speed was re-established to normality indices, as shown in Figure 15.

This analysis was performed 3 months after the event that caused the increase in vibration. The increase in vibration rates was generated by another failure, which, in this case, was related to the frequency inverter. This type of correlation, without access to the data displayed by the sensor historically, would be more difficult to assimilate.

After 23 March, engines 1 and 2 had their anti-vibration systems changed and started to operate within the standard operating range.

### Impacts of Sensor Selection

The selection of a smart sensor for this operation directly affects operational reliability issues, as it is capable of supporting different performances in relation to reducing system failure risks through reliability studies and a capacity-dependent predictive maintenance system for data acquisition according to the selected sensor.

To demonstrate a coherent selection of criteria and selection method, an FMEA (Failure Modes and Effects Analysis) [94] was performed in relation to the risks present in the operation of the electric motor. This FMEA, performed before and after the sensors were installed, worked by comparing the risk exposure level of the equipment before and after the sensors were installed.

In order to eliminate or mitigate the effects, the FMEA was able to select the ideal version for the design and development of the document base. This will support future projects to reduce risks associated with faulty systems/products used by customers [94].

The FMEA team, composed of reliability engineers, technicians, and operators responsible for the operation, analyzes every fault mode and determines the potential effects of the malfunction if they occur using the failure modes listed in the FMEA Work-sheet. The next stage is to assign rankings from 1 to 10 for the severity (S), occurrence (O), and detection (D). Level 1 relates to the most desirable situation, whereas level 10 calls attention to the fact that an issue needs to be improved [95].

Severity ranking is an estimation of how serious the effects would be if a given failure did occur [96]. The best way to determine the occurrence ranking is to use process data. This may be in the form of process capacity data or even fault logs. When there are no actual fault data, the team must estimate how often a malfunction can occur. The detection ranking indicates how likely we are to detect a failure or the effect of a failure. This step starts by identifying current controls that may detect a failure or a failure effect [96].

The FMEA team can calculate the Risk Priority Number (RPN): RPN = S × O × D. This value of RPN indicates which the priority risks in relation to the equipment are in a quantitative way.

The FMEA sheet displayed in Table 11 describes the risk analysis scenario in relation to the standard operation of the motor and the main failures observed in a historical database of the last 24 months, while Table 12 describes the updated FMEA, after installing the selected smart sensors on the motors.

After the installation of the sensors, it was possible to identify that there was a significant improvement in the level of reliability of the equipment, since before the installation of the sensors, the FMEA indicated an RPN sum of 1470, and after the installation of the sensors, the RPN sum over the same identified risks was 550.

This 920 RPN reduction, representing a 62% reduction in the RPN in relation to the identified risks, allowed the operation to work with a higher level of equipment reliability, reducing operational losses and increasing the level of service offered to the customer.

In this way, it is possible to state that the selected sensor exerted an influence related to maintenance, since through the FMEA and analysis of the data acquired from the sensor in the parameters related to the operation, there was an improvement in the reliability of the system.

## 7. Conclusions

The application of the AHP-Gaussian model in the selection of a smart sensor for an escalator motor in a subway station allowed a judicious and reliable choice of the most adequate sensor for this specific application to be made.

The AHP-Gaussian model is a decision-making methodology that combines the analytical hierarchy method (AHP) with Gaussian analysis, allowing different evaluation criteria to be weighted objectively and systemically. This makes the sensor selection process more transparent, minimizing subjectivity and the risk of making the wrong decision.

By using this methodology, it was possible to select the most appropriate sensor for the application in question, considering important criteria such as accuracy, reliability, durability, cost, and ease of installation. Choosing the most suitable sensor can contribute significantly to the efficiency and safety of the equipment, in addition to saving resources in the long term.

The application of the smart sensor selected in the motor of the escalator at the train station also allowed for the continuous, real-time monitoring of the equipment’s operation, allowing for the early detection of possible failures and more efficient preventive maintenance. This contributes to reducing equipment downtime, reducing costs, and, most importantly, increasing user safety.

In summary, the use of the AHP-Gaussian model in the selection of the smart sensor for the escalator motor of a subway station, combined with its practical application, allowed for a careful choice of the most suitable sensor to be made and a significant improvement in the efficiency, safety, and reliability of the equipment.

### Future Scope

Considering future work and future applications, it would be interesting to study the application of other multicriteria decision support methods (MCDMs). Several methods can be evaluated regarding the possibility of application for the study carried out, such as the hybrid methods WSM, WPM, and WASPAS presented by [97], or adjusting the choice of the best method, as proposed by Ref. [98].

It is possible to investigate the recent fuzzy approach through PyIFDM [99], Fuzzy TO-PSIS [100], F-SWARA’B, and F-CoCoSo’B, [101] Ct-SFS [102], and SFAHP [103] for similar applications. Other recent methods or frameworks such as SVN-COPRAS, SVN-TOPSIS [104], IVIF MULTIMOORACOPRAS [105], AHP-COPRAS [106], SAPEVO-H^2^ [107], ECOPRAS [108], SPC [109], TOPANSIS [110], MPSI–MARA [111], MEREC-MARCOS [112], or PROMETHEE-SAPEVO-M1 [113] are examples of models that could be evaluated for adherence to this application, seeking comparative results and different analyses.

## Figures and Tables

**Figure 1 sensors-23-04131-f001:**
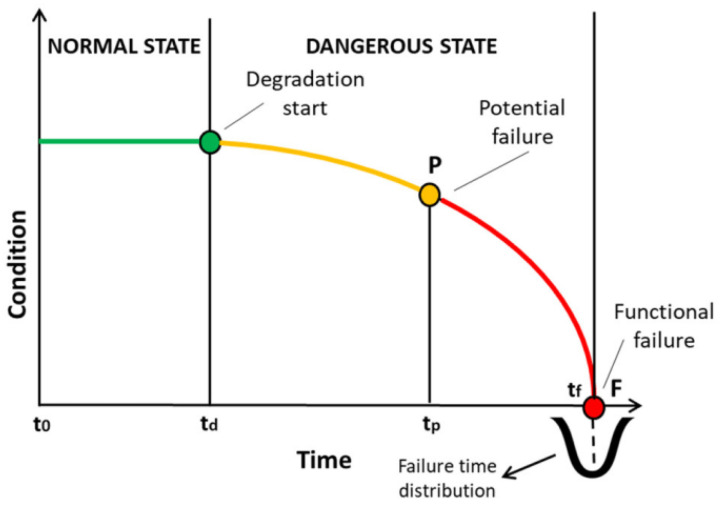
P-F curve.

**Figure 2 sensors-23-04131-f002:**
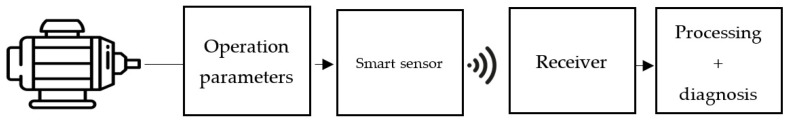
Electric motor monitoring via smart sensor diagram.

**Figure 3 sensors-23-04131-f003:**
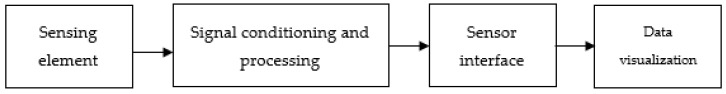
Traditional sensors logic diagram.

**Figure 4 sensors-23-04131-f004:**
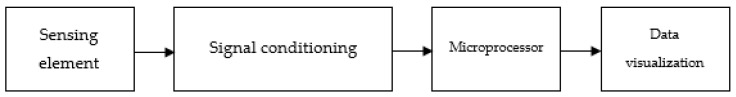
Smart sensor logic diagram.

**Figure 5 sensors-23-04131-f005:**
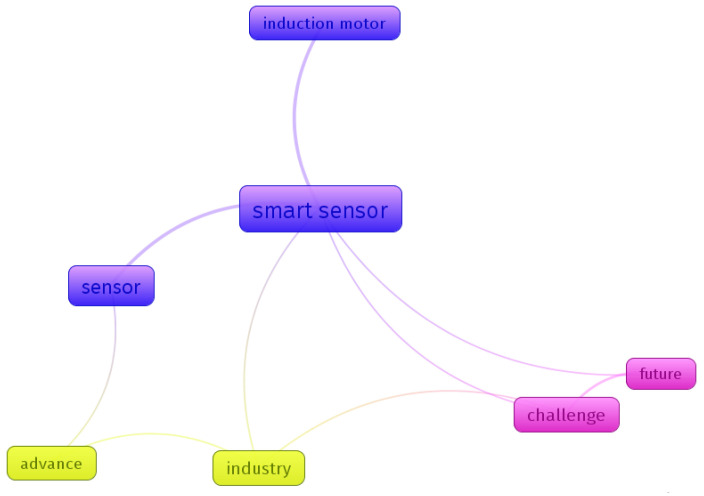
VOS viewer network connection.

**Figure 6 sensors-23-04131-f006:**
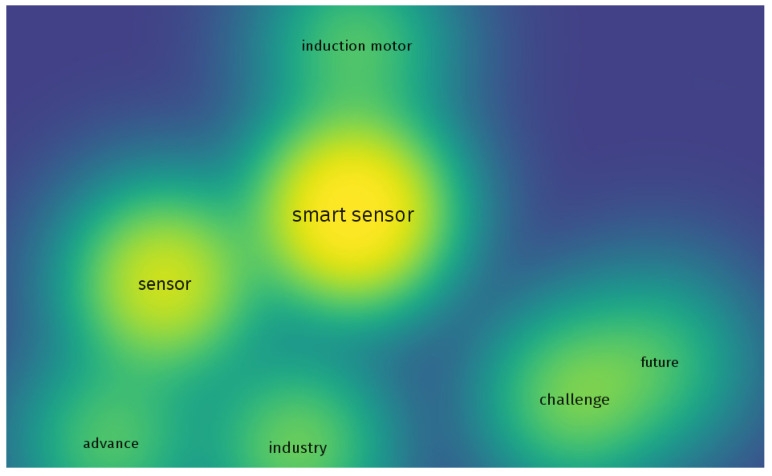
VOS viewer network connection density.

**Figure 7 sensors-23-04131-f007:**
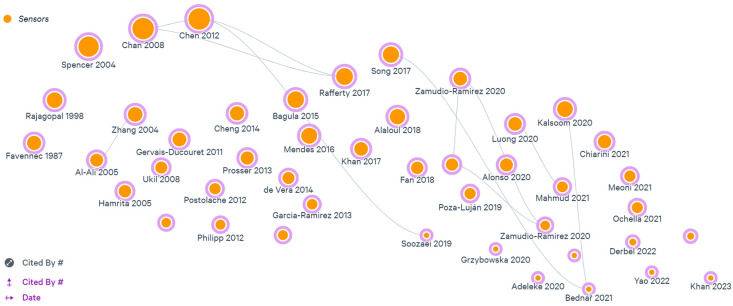
Litmap diagram of citations and years between articles.

**Figure 8 sensors-23-04131-f008:**
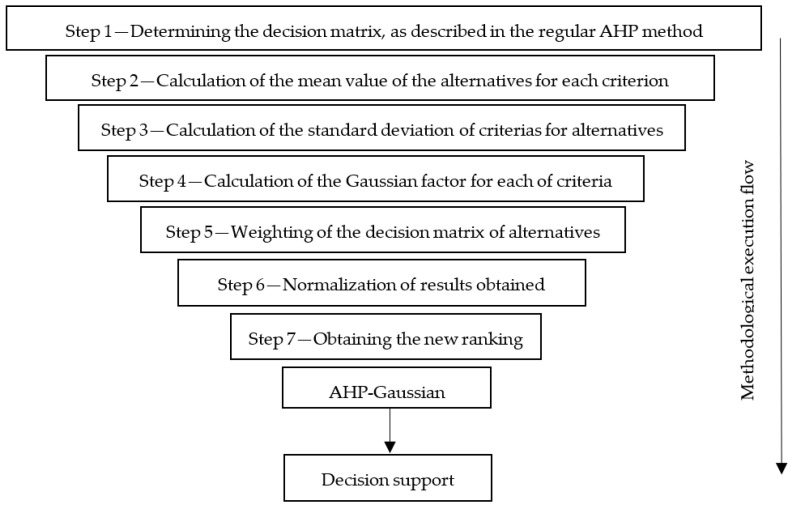
Steps for AHP-Gaussian.

**Figure 9 sensors-23-04131-f009:**
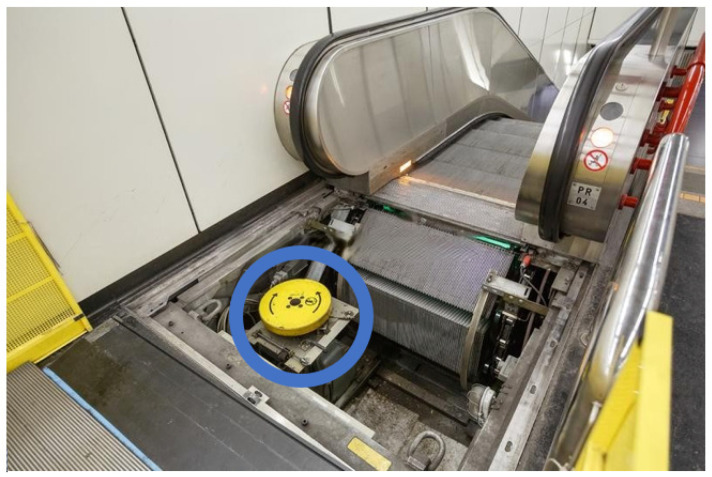
An electric motor in a subway escalator.

**Figure 10 sensors-23-04131-f010:**
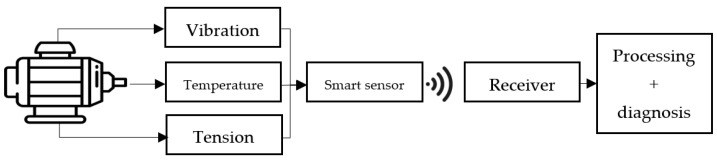
Logic diagram for monitoring an electric motor by the smart sensor.

**Figure 11 sensors-23-04131-f011:**
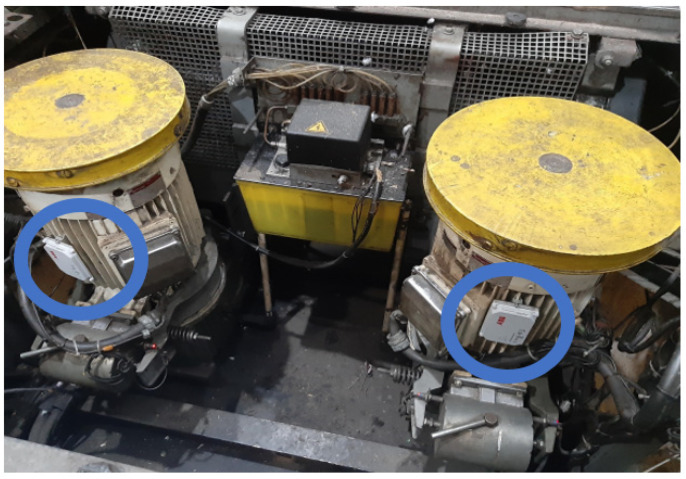
The electric motors of escalator are monitored by smart sensors.

**Figure 12 sensors-23-04131-f012:**
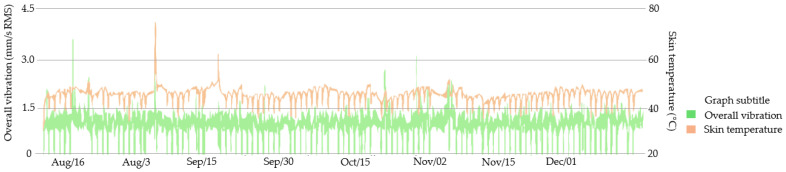
Motor 1 previous monitoring by the smart sensor.

**Figure 13 sensors-23-04131-f013:**
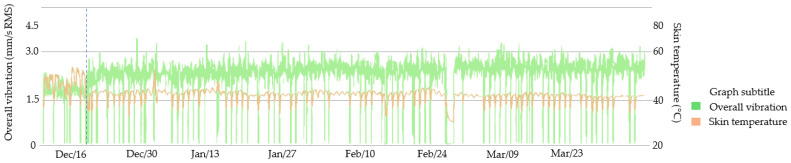
Motor 1 monitoring by the smart sensor.

**Figure 14 sensors-23-04131-f014:**
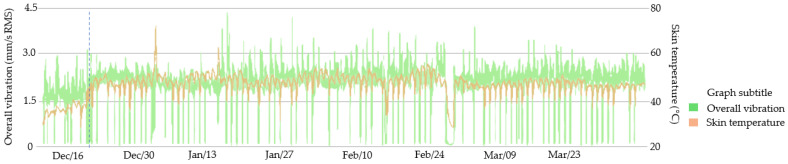
Motor 2 monitoring by the smart sensor.

**Figure 15 sensors-23-04131-f015:**
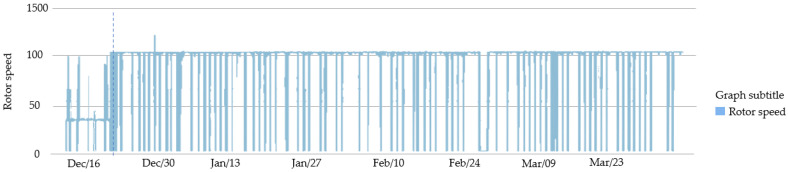
Frequency inverter monitoring by the smart sensor.

**Table 1 sensors-23-04131-t001:** Functions of a smart sensor.

Characteristics	Description
Multi-functionality	Smart sensors are capable of not only detecting and measuring physical variables, but also processing, storing, and transmitting data.
Connectivity	Smart sensors can communicate wirelessly with other devices and systems, allowing for real-time data exchange and control.
Autonomy	Smart sensors can make decisions based on their measurements, without the need for human intervention.
Intelligence	Smart sensors use algorithms and machine learning techniques used to analyze data, identify patterns, and make predictions.
Integration	Smart sensors can be used to integrate systems that were previously installed.
Miniaturization	Smart sensors are often designed to be small and compact, making them suitable for use in a variety of applications and environments.
Cost-effectiveness	Smart sensors offer a cost-effective solution for monitoring and controlling physical processes, compared to traditional approaches that require multiple sensors, data loggers, and other hardware.

**Table 2 sensors-23-04131-t002:** Steps of literature review.

Literature Review Step	Description	SCOPUS	Science Direct
1	Initial search with “smart sensors” AND “Selection” terms.	252	19
2	Limitation to works written in English.	241	19
3	Limitation of the search to specific keywords, aligned with the objective of the work.	120	14
4	Removal of duplicate works in scientific databases.	110	14
5	Acquisition of works for complete reading.	89	10
6	Complete reading.	84	8
Works used as base for literature review		92	

**Table 3 sensors-23-04131-t003:** Theoretical decision parameters of smart sensors.

Parametervalue	Meaning	Description
1	Equal relevance	The two alternatives contribute equally to the goal.
3	Moderate relevance	Experience and judgment slightly favor one activity over another.
5	Strong relevance	Experience and judgment strongly favor one activity over another.
7	Very strong relevance	One alternative is strongly favored over another.
9	Extreme relevance	The evidence is in favor of one alternative over another (to the greatest extent possible).
2,4,5,8	Intermediate values	Used to express preferences that are between the references above.

**Table 4 sensors-23-04131-t004:** Parameters decision of smart sensors.

#	Parameter	Description
C1	Temperature range	This refers to the temperature range that the sensor can operate in, and it is important because it affects the accuracy and reliability of the sensor readings. A wider temperature range means that the sensor is more versatile and can be used in a wider range of environments.
C2	Vibration range	This refers to the range of vibrations that the sensor can withstand and still produce accurate readings. A higher vibration range means that the sensor is more durable and can be used in harsh environments.
C3	Weight	The weight of the sensor is an important factor to consider, especially if the motor is going to be mounted in a limited space or needs to be portable. A lighter sensor is more convenient and easier to install.
C4	Distance of communication	This refers to the maximum distance that the sensor can transmit data to a receiving device. This is important to consider if the motor is located far from the data acquisition system. A longer communication distance means that the sensor can be used in larger installations.
C5	Max electric power (Hp)	Electric power is an important parameter to measure in an electric motor because it provides a measure of the rate at which work is being performed. By measuring the electric power, you can determine how much electrical energy is being converted into useful mechanical energy and assess the efficiency of the motor.
C6	Data Traffic Speed	The data traffic speed of the sensor refers to the speed at which it can transmit data. This is important because the speed at which the sensor can transmit data has a direct impact on the real-time monitoring and control of the motor. If the data traffic speed is slow, there may be a delay in the transmission of data, leading to less accurate and less responsive control of the motor.
C7	Acquisition cost	The cost of acquiring the sensor is an important factor to consider when evaluating different options. A lower cost may mean that the sensor is less sophisticated or has lower performance compared to sensors that are more expensive.

**Table 5 sensors-23-04131-t005:** Decision parameters of smart sensors.

#	Sensor Name Model	Image of Sensor	Max Temperature Work (°C)	Max Vibration Range (KHz)	Weight (g)	Max Distance Communication (m)	Max Electric Power (Hp)	Max Data Traffic Speed (GHz)	Average Acquisition Cost (USD *)
A1	ABB ability smart sensor	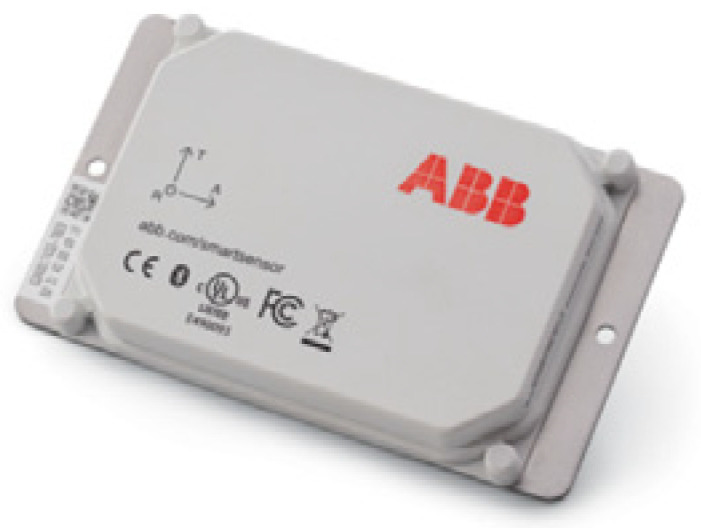	115	13.3	250	50	800	2.4	380
A2	WEG motor scan	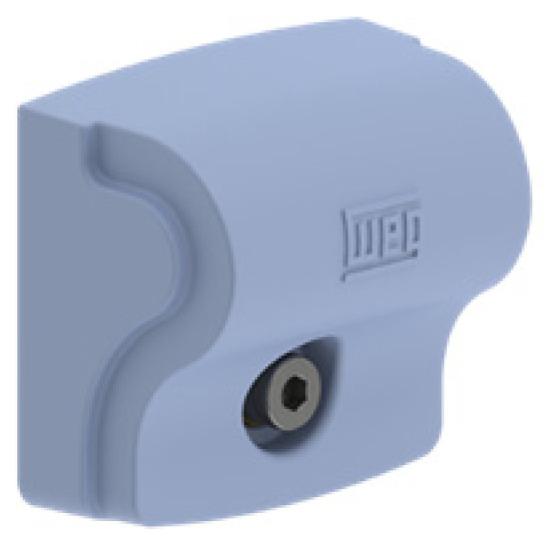	80	13	277	70	700	2.4	290
A3	Hoyer smart sensor	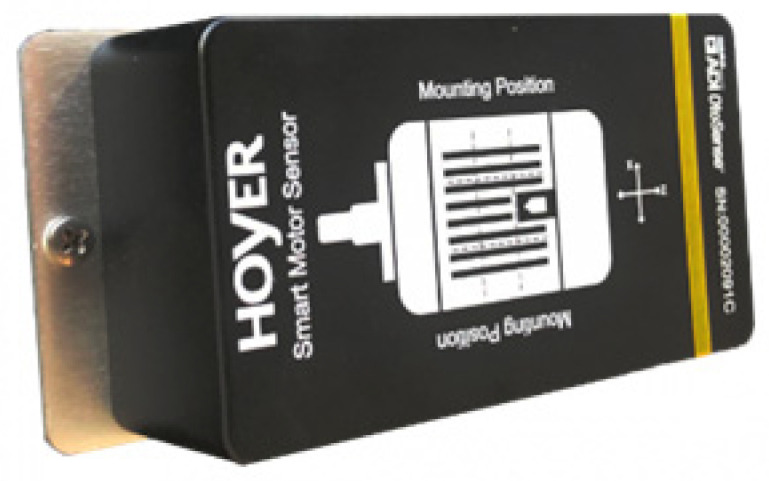	80	5.1	500	50	700	2.4	560
A4	Dynamox Dynalogger	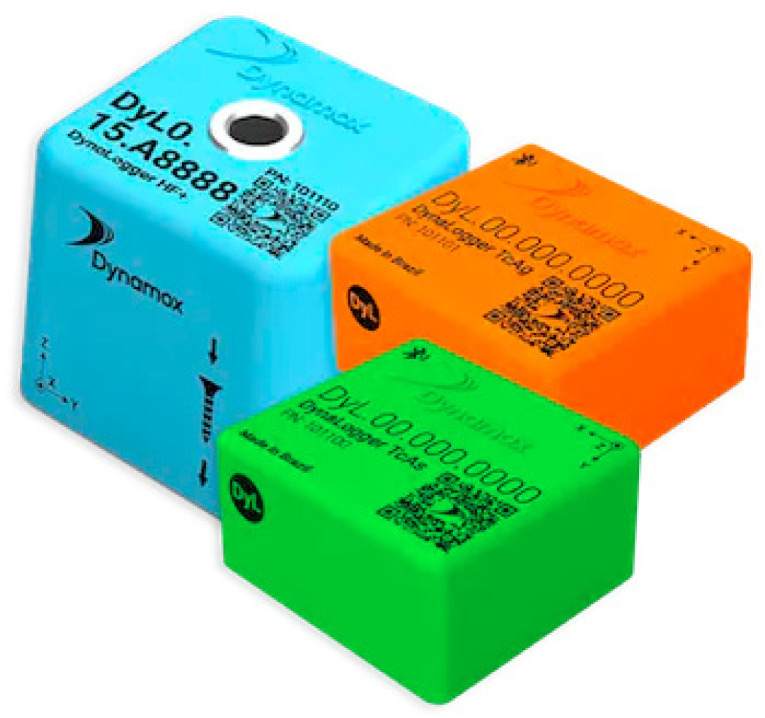	100	2.5	320	60	1000	5.3	350

* Source: [88,89,90,91]. Quotation USD-BRL 7 April 2023.

**Table 6 sensors-23-04131-t006:** Matrix of decision—integral.

#	Alternatives	C1	C2	C3	C4	C5	C6	C7
		Max	Max	Min	Max	Max	Max	Min
		Temperature Work (°C)	Vibration Range (KHz)	Weight (g)	Distance Communication (m)	Electric Power (Hp)	Data Traffic Speed (GHz)	Acquisition Cost (USD)
A1	ABB Ability Sensors	115	13.3	250	50	800	2.4	380
A2	WEG motor Scan	80	13	277	70	700	2.4	290
A3	Hoyer Smart sensor	80	5.1	500	50	700	2.4	560
A4	Dynamox Dynalogger Sensor	100	2.5	320	60	1000	5.3	350

**Table 7 sensors-23-04131-t007:** Matrix of decision—Normalized—Step 1.

#	Alternatives	C1	C2	C3	C4	C5	C6	C7
		Max	Max	Min	Max	Max	Max	Min
		Temperature Work (°C)	Vibration Range (KHz)	Weight (g)	Distance Communication (m)	Electric Power (Hp)	Data Traffic Speed (GHz)	Acquisition Cost (USD)
A1	ABB Ability Sensors	0.3067	0.3923	0.3141	0.2174	0.2500	0.1920	0.2454
A2	WEG motor Scan	0.2133	0.3835	0.2835	0.3043	0.2188	0.1920	0.3215
A3	Hoyer Smart sensor	0.2133	0.1504	0.1570	0.2174	0.2188	0.1920	0.1665
A4	Dynamox Dynalogger Sensor	0.2667	0.0737	0.2454	0.2609	0.3125	0.4240	0.2664

**Table 8 sensors-23-04131-t008:** Average criteria—Standard deviation—Gaussian factor—Step 2, 3 and 4.

AHP-G Parameters	C1	C2	C3	C4	C5	C6	C7
	Temperature Work (°C)	Vibration Range (KHz)	Weight (g)	Distance Communication (m)	Electric Power (Hp)	Data Traffic Speed (GHz)	Acquisition Cost (USD)
Average	0.2500	0.2500	0.2500	0.2500	0.2500	0.2500	0.2500
Standard deviation	0.0454	0.1623	0.0680	0.0416	0.0442	0.1160	0.0642
Gaussian factor	0.1815	0.6493	0.2722	0.1665	0.1768	0.4640	0.2569
Gaussian factor normalized	0.0838	0.2996	0.1256	0.0768	0s.0816	0.2141	0.1185

**Table 9 sensors-23-04131-t009:** Average criteria—standard deviation—Gaussian factor—Step 5.

#	Alternatives	C1	C2	C3	C4	C5	C6	C7
		Max	Max	Min	Max	Max	Max	Min
		Temperature Work (°C)	Vibration Range (KHz)	Weight (g)	Distance Communication (m)	Electric Power (Hp)	Data Traffic Speed (GHz)	Acquisition Cost (USD)
A1	ABB Ability Sensors	0.3067	0.3923	0.3141	0.2174	0.2500	0.1920	0.3067
A2	WEG motor Scan	0.2133	0.3835	0.2835	0.3043	0.2188	0.1920	0.2133
A3	Hoyer Smart sensor	0.2133	0.1504	0.1570	0.2174	0.2188	0.1920	0.2133
A4	Dynamox Dynalogger Sensor	0.2667	0.0737	0.2454	0.2609	0.3125	0.4240	0.2667

**Table 10 sensors-23-04131-t010:** Average criteria—standard deviation—Gaussian factor.

#	Alternatives	$\sum G_{n}*A_{i}C_{j}$	AHP-G	Ranking
A1	ABB Ability Sensors	0.2900	0.145028	1
A2	WEG motor Scan	0.2888	0.406709	2
A3	Hoyer Smart sensor	0.1781	0.222807	4
A4	Dynamox Dynalogger Sensor	0.2431	0.225456	3

**Table 11 sensors-23-04131-t011:** FMEA before selected smart sensor installation.

FMEA										
#	Failure		Effect	S	Cause	O	Controls		D	RPN
	Function	Failure Mode					Prevention	Detection		
1	Coupling the motor to the escalator	Motor decoupling	Escalator shaking	9	High level of motor vibration	5	Maintenance every 4 months	External inspection every week	5	225
2	Regular operation	Motor heating	Reduced motor shelf life	6	Changing the energy level and effort of the motor to operate	6	Without prevention	During corrective maintenance	8	288
2.1	Regular operation	Changing operating speed	Unsafety transport of users	8	Logic controller and frequency inverter malfunction	6	Without prevention	During corrective maintenance	8	384
2.2	Regular operation	Transmission chain breakage	Loss of operating capacity	9	High level of motor vibration	5	Maintenance every 4 months	External inspection every week	4	180
2.3	Regular operation	Misalignment of the steps	Loss of operating capacity	9	Clearance on the rails of the steps	5	Maintenance every 2 months	External inspection every week	5	225
2.4	Regular operation	High level of noise during operation	User discomfort	7	High level of temperature and vibration in the transmission system	8	Without prevention	External inspection every week	3	168

**Table 12 sensors-23-04131-t012:** FMEA after selected smart sensor installation.

FMEA										
#	Failure		Effect	S	Cause	O	Controls		D	RPN
	Function	Failure Mode					Prevention	Detection		
1	Coupling the motor to the escalator	Motor decoupling	Escalator shaking	9	High level of motor vibration	5	Full-time sensing and monitoring of vibration indices	Analysis software integrated into sensors	2	90
2	Regular operation	Motor heating	Reduced Motor shelf life	6	Changing the energy level and effort of the motor to operate	6	Full-time sensing and monitoring of temperature indices	Analysis software integrated into sensors	2	72
2.1	Regular operation	Changing operating speed	Unsafety transport of users	8	Logic controller and frequency inverter malfunction	6	Motor speed sensing and monitoring	Analysis software integrated into sensors	2	96
2.2	Regular operation	Transmission chain breakage	Loss of operating capacity	9	High level of motor vibration	5	Full-time sensing and monitoring of vibration indices	Analysis software integrated into sensors	2	90
2.3	Regular operation	Misalignment of the steps	Loss of operating capacity	9	Clearance on the rails of the steps	5	Full-time sensing and monitoring of vibration indices	Analysis software integrated into sensors	2	90
2.4	Regular operation	High level of noise during operation	User discomfort	7	High level of temperature and vibration in the transmission system	8	Full-time sensing and monitoring of vibration and temperature indices	Analysis software integrated into sensors	2	112

## Data Availability

Not applicable.

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
