# Peer review of "Evaluation of Smart Sensors for Subway Electric Motor Escalators through AHP-Gaussian Method"

_sensors, 2023, doi:10.3390/s23084131_

Round 1
Reviewer 1 Report
In this manuscript, authors proposed the use of the AHP-Gaussian method to support the selection of a smart sensor installation for an electric motor used in an escalator in a subway station. Authors also claim that the proposed AHP-Gaussian method proved to be 23 an effective approach for selecting a smart sensor for an electric motor used in an escalator in a 24 subway station However, I have few observations:
1. Why the authors have chosen only AHP-Gaussian method why not other methods like fuzzy topsis or best worst method which are quite popular nowadays Authors are required to elaborate more clearly on it and on the novelty of their work..
2. Authors are also advised to add the future scope in the conclusion section as well. Authors are required to go through recent references related to various recently developed in MCDM techniques and their applications in achieving the SDG to make the reference list exhaustive. For example:
(i) Kumar, A., et al. (2022). Multi-Criteria Decision-Making Techniques for Complex Decision Making Problems. Mathematics in Engineering, Science & Aerospace (MESA), 13(2).
3. Authors are required to add a subsection clearly articulating the main limitations and wider applicability of their model.
4. However, the English of the manuscript is readable though I suggest proofreading of the manuscript carefully for grammatical errors.
5. Recheck all the captions of figures, equations and tables.
Author Response
Dear reviewer,
thank you for your great review.
Please see the attachment, showing the changes in red.
Summarizing the point adjusted points:
1 - Final of chapter 3 and begin of chapter 4.;
2- Added the section 7.1;
3- Added the section 4.1;
4- English checked;
5- Captions checked.
Best regards.

Reviewer 2 Report
The article requires corrections.
Editorial corrections
The sensor data in Table 5 and Table 6 do not match.
Figure 3 and Figure 4 should be removed as they do not provide new information or contribute to the understanding of the content of the article.
Chapter 3 Literature Review is too extensive and deals with issues not related to the methods used in the article. The aim of the article, in my opinion, is not related to scientific research on systems referred by the authors in the chapter such as: intelligent control, sensors networking, internet of things, automation, decision support system, maintenance or microsensors, but deals with the decision-making process concerning the selection of a monitoring system component, which is one of several smart sensors available on the market for monitoring the operation of electric motors.
Comments on the content of the article
The article should be expanded to address issues related to the article's objective of decision-making process. The article presents only one arbitrarily selected method to support the decision making process. Parameters were arbitrarily selected without any analysis of their degree of importance on maintenance-related issues and the results were presented in the form of recorded data from a selected sensor without any analysis of how the selected criteria and method affect maintenance-related system parameters.
Author Response
Dear reviewer,
thank you for your great review.
Please see the attachment, showing the changes in red.
Summarizing the point adjusted points:
1- Fixed the tables 5 and 6;
2- Removed the Figure 3 and Figure 4;
3- Adjustments done in Chapter 3 and Chapter 4.;
4- Added the section 6.1 and 7.1.
Best regards.

Round 2
Reviewer 1 Report
addressed all suggested corrections
Author Response
Dear reviewer,
thank you for your great review.
Best Regards.

Reviewer 2 Report
After making editorial corrections, the article is ready for publication.
My comments from the previous review regarding the analysis of sensor influence on maintenance-related issues concerned the analysis of the impact of the acquired sensor data on parameters related to system operation and reliability. The information contained in Chapter 6.1 and Table 11 does not respond to my comment and does not add anything to the article compared to the previous version.
In my opinion, carrying out such an analysis, even with the use of the FMEA (Failure Mode and Effect Analysis) method, concerning the reduction of risks related to system failure, would make it possible to demonstrate the correct selection of criteria and selection method.
Author Response
Dear reviewer,
thank you for your great review.
Please see the attachment. The changes are in red.
Just a summary of the changes:
1) I fixed the section 6.1 and table 11, following your recommendations;
2) Added de table 12.
Best Regards.
